# Expression and Purification of Recombinant SARS-CoV-2 Accessory Protein ORF7a and Functional Analysis of Its Role in Up-Regulating Cytokine Production

**Dan Chen [1], Zhenhua Zheng [2] and Zhenggang Han [1],***

1    School of Life Science and Technology, Wuhan Polytechnic University, Wuhan 430023, China
2    CAS Key Laboratory of Special Pathogens and Biosafety, Center for Emerging Infectious Diseases, Wuhan Institute of Virology, Chinese Academy of Sciences, Wuhan 430071, China
*    Correspondence: zhengganghan@whpu.edu.cn

**Abstract:** The severity of coronavirus disease 2019 is closely linked to dysregulated immune responses. The search for viral proteins associated with immune regulation in severe acute respiratory syndrome coronavirus 2 (SARS-CoV-2) is critical to reveal the pathogenicity of the virus. In this study, accessory proteins ORF7a (referred to as ORF7a-1 and ORF7a-2, respectively) from two SARS-related coronaviruses, severe acute respiratory syndrome coronavirus (SARS-CoV) and SARS-CoV-2, were produced through the denaturing and refolding of inclusion body proteins. The recombinant protein was incubated with alveolar epithelial cells, and the transcription and expression levels of major cytokines were determined by reverse transcription-quantitative PCR and enzyme-linked immunosorbent assay. SARS-related coronavirus ORF7a can up-regulate the transcription and expression of interleukin-6, C-C motif chemokine ligand 8, interferon α, and interferon β. The results also indicated that the two highly conserved ORF7a had certain differences in promoting the transcription and expression of cytokines. The study showed that ORF7a is a virus-encoded immune regulator by alveolar epithelial cells that plays an important role in the pathogenicity of SARS-related coronaviruses.

**Keywords:** COVID-19; SARS-CoV-2; accessory protein; open reading frame 7a; cytokine release syndrome

## 1. Introduction

The 2019 novel coronavirus, severe acute respiratory syndrome coronavirus 2 (SARS-CoV-2), is the third human coronavirus with lethal consequences after severe acute respiratory syndrome coronavirus (SARS-CoV) and Middle East respiratory syndrome coronavirus (MERS-CoV) [1]. Since the beginning of the SARS-CoV-2 pandemic, more than 600 million cases of infection and more than 6.4 million deaths have been reported worldwide (WHO statistics on 2 September). The virus had a devastating impact on global public health and socioeconomic development. The clinical course and severity of coronavirus disease 2019 (COVID-19) manifestations vary widely. Most people infected with SARS-CoV-2 develop mild upper respiratory symptoms or even asymptomatic infection. However, the elderly and people with underlying diseases, such as cardiovascular disease, diabetes, and chronic respiratory disease, are prone to develop severe disease [2]. To date, the pathogenicity of SARS-CoV-2, SARS-CoV, and MERS-CoV have not been fully understood. However, an increasing number of studies have shown that severe COVID-19 patients exhibited excessive inflammation caused by massive cytokine/chemokine production, termed cytokine release syndrome (CRS) [3]. CRS has attracted much attention due to its potential association with severe infections by viruses such as SARS-CoV [4], MERS-CoV [5], influenza A virus [6], and Hantavirus [7].

Cytokines are membrane-bound or secreted glycoproteins that control the infection and limit the spread of viruses [8]. However, the excessive or even uncontrolled release

of proinflammatory cytokines and chemokines, such as tumor necrosis factor-alpha (TNF-α), interleukin-1 beta (IL-1β), interleukin-6 (IL-6), interleukin-8 (IL-8), and interleukin-17 (IL-17), in severe COVID-19 patients is associated with extensive lung injury, acute respiratory distress syndrome, and eventual death [9,10]. Particularly, the concentration of IL-6 in the serum of infected patients was significantly related to the clinical manifestations and mortality of patients, so it received more attention [11]. This association prompted the clinical use of IL-6 levels as a predictor of possible progression to more severe disease [12] and the use of anti-IL-6 therapy for the treatment of patients with moderate to severe COVID-19 [13].

Coronaviruses usually encode a variable number of accessory proteins. It is generally believed that these accessory proteins are not essential for viral replication, but some accessory proteins have been shown to be involved in virus–host interactions and play an important role in the pathogenesis of coronaviruses [14,15]. The SARS-CoV-2 genome encodes 31 proteins, including 16 nonstructural proteins (nsp1-16) involved in genome replication and early transcriptional regulation, four structural proteins (spike, envelope, membrane, and nucleocapsid proteins), and 11 accessory proteins (ORF3a, ORF3b, ORF3c, ORF3d, ORF6, ORF7a, ORF7b, ORF8, ORF9b, ORF9c, and ORF10) [16]. Several SARS-CoV-2 accessory proteins have been identified to participate in the pathological inflammatory response of COVID-19 and played an important role in the pathogenicity of the virus [17–19].

Both SARS-CoV and SARS-CoV-2 encode the accessory protein ORF7a. ORF7a is a small type-I transmembrane protein consisting of an N-terminal signal peptide, an immunoglobulin-like ectodomain, a transmembrane region, and a C-terminal tail [15,18]. ORF7a from SARS-CoV (ORF7a-1) is 86% identical in amino acid sequence to ORF7a from SARS-CoV-2 (ORF7a-2). Earlier studies found that ORF7a-1 had the ability to induce the excessive production of proinflammatory cytokines in host cells [20]. Due to the high similarity in amino acid sequences between the two ORF7a encoded by the two viruses, it is likely that ORF7a-2 also has the capability to stimulate host cells to produce proinflammatory cytokines. Researchers either used transient transfection to express ORF7a inside lung or bronchial epithelial cells [21] or used recombinant ORF7a ectodomains to stimulate immune cells extracellularly [22]. In contrast, we employed a way that the recombinant protein of ORF7a ectodomain was used to stimulate alveolar basal epithelial cells extracellularly. This combination of the experimental approach and cell type has not been performed in previous studies. In this study, the N-terminal ectodomains of the two ORF7a proteins were obtained by inclusion body refolding. Using high-pure recombinant ORF7a to stimulate human alveolar epithelial cells, it was demonstrated that both ORF7a-1 and ORF7a-2 have the ability to induce the up-regulation of a series of cytokines extracellularly in lung epithelial cells, but albeit with differences. The possible reasons for the functional differences between ORF7a-1 and ORF7a-2 were explored through three-dimensional structural analysis.

## 2. Materials and Methods

### 2.1. Materials

Genes encoding the ectodomains (amino acid residues ranging from 16–94) of SARS-CoV ORF7a-1 (SARS-CoV isolate Tor2, Genbank: NC_004718.3) and SARS-CoV-2 ORF7a-2 (SARS-CoV-2 isolate Wuhan-Hu-1, Genbank: NC_045512.2) were synthesized by Nanjing GenScript (Nanjing, China). Rabbit anti-ORF7a-2 polyclonal antibody was kindly provided by Dr. Zhou Peng, Wuhan Institute of Virology. Human alveolar epithelial cells were preserved in our laboratory and cultured with DMEM medium (Fisher Scientific, Shanghai, China) containing 10% fetal bovine serum (Fisher Scientific, Shanghai, China) at 37 °C in a 5% $CO_2$ incubator [23]. The human cytokine detection kits were purchased from Wuhan Saipei Biotechnology (Wuhan, China).

## 2.2. Methods

### 2.2.1. Expression and Purification of the Ectodomains of ORF7a-1 and ORF7a-2

The genes of the ectodomains of ORF7a-1 and ORF7a-2 were inserted into an expression cassette on the *Escherichia coli* expression vector pET-28a by *Eco*RI and *Not*I restriction endonucleases (New England Biolabs, Beijing, China). The recombinant plasmids (Supplemental Figure S1) were transformed into *E. coli* BL21 (DE3) cells (Fisher Scientific, Shanghai, China), respectively. The *E. coli* containing the recombinant plasmids were inoculated in 1 L of ZYM 5052 autoinduction medium, respectively [24]. After culturing for 6 h at 37 °C, the cells were collected by centrifugation at 5000 rpm for 30 min at 4 °C. The cells were resuspended in 20 mL of lysis buffer (500 mM NaCl, 100 mM Tris-HCl, pH 8.5, 1% Triton-X 100) and sonicated in an ice bath for 40 min. After centrifugation at 10,000 rpm for 30 min at room temperature, the cell lysate pellet was collected. The pellet was fully dissolved in 20 mL of inclusion body denaturation solution (500 mM NaCl, 100 mM Tris-HCl, pH 8.5, 6 M guanidine hydrochloride) and left at room temperature overnight. The undissolved cell debris was removed by centrifugation at 13,000 rpm for 30 min at 4 °C. The solubilized inclusion body solution was dropwised to 10 volumes of inclusion body renaturation solution (500 mM NaCl, 100 mM Tris-HCl, pH 8.5, 1 M arginine, 5 mM reduced glutathione and 500 μM oxidized glutathione) and stirred to mix. Refolding was performed overnight at 4 °C. The renatured protein was dialyzed against 2 L of nickel column equilibration solution (500 mM NaCl, 50 mM Tris-HCl, pH 8.5). The dialyzed renatured proteins were then loaded onto a HisTrap prepacked nickel column (GE Healthcare Life Sciences, Beijing, China). The recombinant proteins were eluted from the column using a linear concentration of imidazole (10–500 mM). The eluted proteins were dialyzed against an equilibrium solution (phosphate-buffered saline) used for size-exclusion chromatography. After purification using the Superdex 75 column (GE Healthcare Life Sciences, Beijing, China), the recombinant ORF7a-1 and ORF7a-2 ectodomain proteins were purified. After appropriate concentration by ultrafiltration, the protein concentration was determined using the Bradford method.

### 2.2.2. Immunoblotting

The recombinant ORF7a-1 and ORF7a-2 proteins (approximately 3 μg) were separated by 15% sodium dodecyl sulfate–polyacrylamide gel electrophoresis (SDS-PAGE), and were transferred to the PVDF membrane (Millipore, Shanghai, China) (200 V, 400 mA). After blocking with TBST buffer (10 mM Tris–HCl, 150 mM NaCl, 0.05% Tween-20, and 5% nonfat dry milk) for 1 h at room temperature, the polyvinylidene fluoride (PVDF) membrane was incubated with ORF7a-2 polyclonal antibody at 4 °C overnight. The membrane was washed 3 times with TBST and then incubated with a peroxidase-labeled secondary antibody (primary concentration: approximately 2 mg/mL, 1:10,000 dilution) for an additional 1 h at room temperature. After washing 3 times with TBST, the protein bands were detected using the ECL detection system and photographed (Bio-Rad Laboratories, Shanghai, China).

### 2.2.3. Cell Experiments

The A549 cells (Fisher Scientific, Shanghai, China) were passaged into 6-well plates. When the cells had grown to approximately 80% density, 20 μg of ORF7a-1 and ORF7a-2 ectodomain proteins were added to 2 mL of cell-culture supernatant, respectively. After a 2 h incubation at 37 °C, the medium was changed to fresh DMEM containing 2% serum.

### 2.2.4. Reverse Transcription and Quantitative PCR (RT-qPCR)

After the recombinant ORF7a proteins were incubated with A549 cells for 48 h, the cells in each well were lysed using TRIzol reagent (Fisher Scientific, Shanghai, China), and the total cellular RNA was extracted. The first-strand cDNA was synthesized using a rapid reverse transcription kit (Tiangen, Beijing, China). The total volume for the qPCR was 20 μL. The system contains 1 μL of cDNA from the reverse transcription reaction (1:2 dilution), 1 μL of upstream and downstream primers, 10 μL of 2 × SYBR Green

Supermix (Bio-Rad Laboratories, Shanghai, China), and 7 µL of $H_2O$. qPCR was performed on the CFX Connect™ real-time PCR detection system (Bio-Rad Laboratories, Shanghai, China). The reaction conditions were 40 cycles, each cycle including two steps of heating to 95 °C and 56 °C. The data analysis was carried out with CFX Manager software (Bio-Rad Laboratories, Shanghai, China). After being normalized to levels of glyceraldehyde 3-phosphate dehydrogenase, the transcriptional levels of the cytokines interleukin-1 alpha (IL-1$\alpha$), IL-1$\beta$, IL-6, IL-8, interferon $\alpha$ (IFN-$\alpha$), interferon $\beta$ (IFN-$\beta$), C-C motif chemokine ligand 2 (CCL-2), C-C motif chemokine ligand 8 (CCL-8), and C-X-C motif chemokine ligand 9 (CXCL-9) in the cells were calculated using the $2^{-\Delta\Delta CT}$ method [25]. The wells without recombinant ORF7a protein were used as the blank controls.

### 2.2.5. ELISA Experiment

After the recombinant ORF7a proteins were incubated with the A549 cells for 48 h, the cell supernatant was collected by centrifugation at 3000 rpm at 4 °C for 10 min to remove cell debris. The levels of IL-1$\alpha$, IL-1$\beta$, IL-6, and IL-8 in the supernatants were determined using the double-antibody sandwich method [26]. The blank wells were set to zero, and the absorbance of each well was measured at a wavelength of 450 nm. The wells without recombinant ORF7a protein treatment were used as the blank controls.

### 2.2.6. Amino Acid and Structural Alignments of ORF7a-1 and ORF7a-2

The amino acid alignment of the ectodomains of ORF7a-1 and ORF7a-2 was performed by the Clustal Omega online program (https://www.ebi.ac.uk/Tools/msa/clustalo/, accessed on 15 August 2022). Structural alignment and root mean squared deviation (RMSD) value calculation were carried out by the structural alignment function of the PyMOL software (Schrödinger). The protein structural figure was prepared with PyMOL.

## 3. Results

### 3.1. Expression, Purification and Identification of the Ectodomains of ORF7a-1 and ORF7a-2

Under the auto-induction condition, after the glucose in the medium is consumed, the expression of exogenous proteins is initiated when the bacteria start to use lactose. Compared with the whole-cell samples cultured for 1 h, the whole-cell samples cultured for 6 h showed a band of a molecular weight smaller than but close to the standard molecular weight protein of 14.3 kDa in SDS-PAGE (Figure 1). The protein size corresponded to the theoretical molecular weight of approximately 13 kDa for the ectodomain of ORF7a. The result indicated that ORF7a-1 and ORF7a-2 proteins are successfully expressed by the auto-induction system. The ectodomains of ORF7a were produced in *E. coli* in the form of inclusion bodies. Therefore, after cell disruption, the centrifugation pellets where the inclusion bodies are located were collected. After fully dissolving the inclusion bodies with a denaturant buffer containing 6 M guanidine hydrochloride, refolding was performed by dropwise dilution. After renaturation, ORF7a-1 and ORF7a-2 produced a large amount of precipitate. However, SDS-PAGE analysis showed that a considerable proportion of the target protein existed in the soluble fraction (Figure 1, lane 3), while the precipitated contained mostly miscellaneous proteins (Figure 1, lane 4).

After dialysis, the refolded ORF7a-1 and ORF7a-2 were purified using a nickel affinity column. The function of nickel column purification was not only to remove the impurity protein but also to concentrate the protein solution (the solution volume after refolding was very large). After further purification by size-exclusion chromatography and proper concentration using an ultrafiltration tube, highly pure ORF7a-1 and ORF7a-2 were obtained, as demonstrated by an approximately single protein band on SDS-PAGE (Figure 1, lane 7). Using bovine serum albumin as the standard, the concentrations of ORF7a-1 and ORF7a-2 measured by the Bradford method were 3.8 and 3.5 mg/mL, respectively.

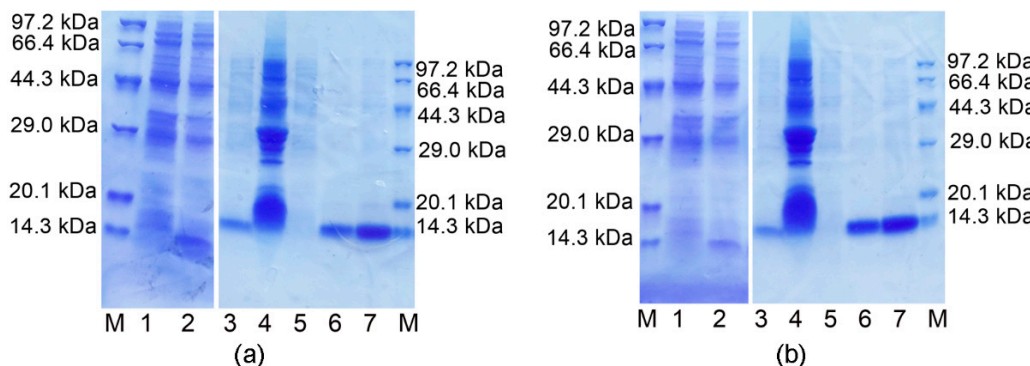

**Figure 1.** Expression and purification of the ectodomains of ORF7a and SDS-PAGE analysis. (**a**) ORF7a-1. (**b**) ORF7a-2. Lane M, protein molecular weight standards; 1, whole cell samples collected from E. coli containing recombinant plasmids after culturing for 1 h; 2, whole cell samples collected from E. coli containing recombinant plasmids after culturing for 6 h; 3, supernatant obtained after refolding; 4, precipitation after refolding; 5, collected solution passing through the nickel column; 6, protein sample obtained after imidazole elution from the nickel column; 7, concentrated protein sample after purification by size-exclusion chromatography.

The purified recombinant ORF7a-1 and ORF7a-2 were transferred onto the PVDF membrane and detected by rabbit anti-ORF7a-2 polyclonal antibodies. The protein band between the 15 kDa and 10 kDa standard proteins was detected for both ORF7a-1 and ORF7a-2 (Figure 2).

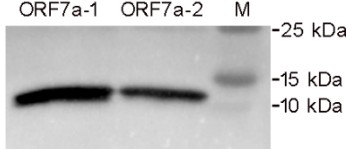

**Figure 2.** ORF7a-1 and ORF7a-2 by Western blot. M, protein molecular weight standards.

*3.2. Induction of Proinflammatory Cytokine Transcription Mediated by Recombinant ORF7a-1 and ORF7a-2*

Human alveolar basal epithelial cells were incubated with the ectodomains of recombinant ORF7a-1 and ORF7a-2 for 48 h, and the transcriptional levels of major cytokines were quantified by RT-qPCR. The transcription of proinflammatory cytokines, interferons, and chemokines was up-regulated to varying degrees in the A549 cells compared with the cells not stimulated with ORF7a (Figure 3). Both ORF7a-1 and ORF7a-2 significantly increased the transcription of IL-6, IFN-$\alpha$, and CCL-8 (Figure 3). However, ORF7a-1 and ORF7a-2 showed certain differences in their ability to promote transcriptional up-regulation of IL-6, IFN-$\beta$, and CCL-8. ORF7a-1 was more effective than ORF7a-2 in up-regulating IL-6, and ORF7a-2 was stronger than ORF7a-1 in up-regulating IFN-$\beta$ and CCL-8. ORF7a-1 and ORF7a-2 did not significantly alter the transcription of IL-1$\alpha$, IL-1$\beta$, IL-8, and CCL-2. The transcription of CXCL-9 was not detected in both the control cells and the cells stimulated with ORF7a protein (data not shown).

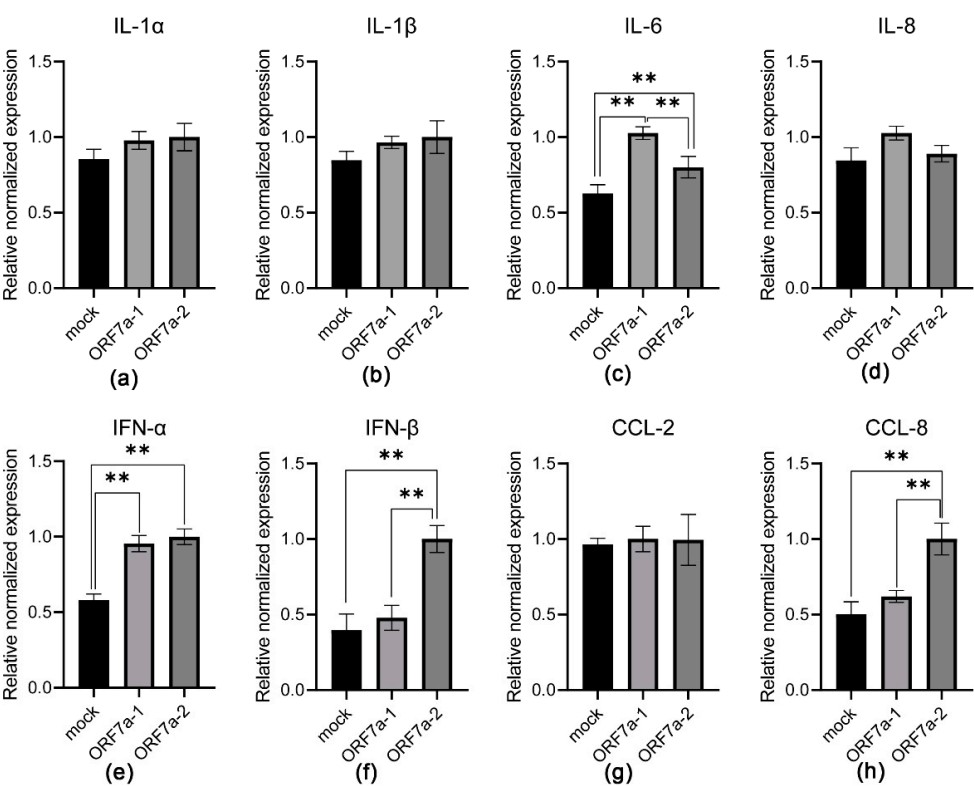

**Figure 3.** The effect of extracellular stimulation of ORF7a protein on the transcription of inflammatory cytokines, interferons, and chemokines in A549 cells. (**a**) IL-1α. (**b**) IL-1β. (**c**) IL-6. (**d**) IL-8. (**e**) IFN-α. (**f**) IFN-β. (**g**) CCL-2. (**h**) CCL-8. Error bars are mean ± standard error of the mean (n = 3). ** $p < 0.01$.

### 3.3. Induction of Proinflammatory Cytokine Expression Mediated by Recombinant ORF7a-1 and ORF7a-2

After stimulating the A549 cells with recombinant ORF7a-1 and ORF7a-2 for 48 h, the expression levels of some proinflammatory cytokines in the cell supernatants were detected by enzyme-linked immunosorbent assay (ELISA) (Figure 4). In the supernatant of A549 cells stimulated by ORF7a, the proinflammatory cytokine IL-6 was significantly up-regulated. The effect of ORF7a-1 in up-regulating IL-6 was stronger than that of ORF7a-2. Similar to that revealed by RT-qPCR, the up-regulation of IL-1α, IL-1β, and IL-8 in lung epithelial cells was not significant after ORF7a stimulation.

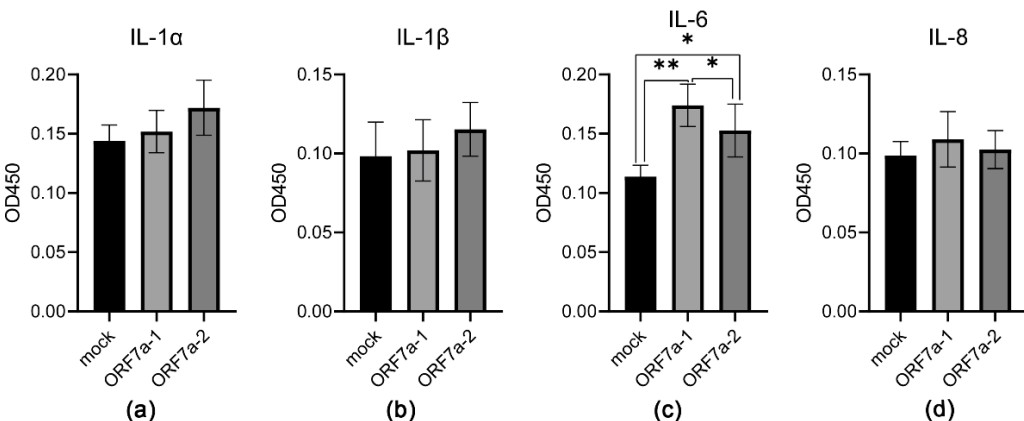

**Figure 4.** Detect the expression levels of cytokines in the supernatant of A549 cells stimulated with recombinant ORF7a by ELISA. (**a**) IL-1α. (**b**) IL-1β. (**c**) IL-6. (**d**) IL-8. Error bars are mean ± standard error of the mean (n = 3). * $p < 0.05$, ** $p < 0.01$.

*3.4. Amino Acid and Structural Comparisons of Ectodomains of ORF7a-1 and ORF7a-2*

The ectodomain of ORF7a-1 and ORF7a-2 have seven differences in amino acid sequence, namely P36S, T59F, H62Q, A68P, T71V, R72K, and T74V (the preceding amino acid is ORF7a-1 followed by ORF7a-2) as revealed by amino acid sequence alignment (Figure 5a). Amino acid residue 72 is an amino acid with similar properties and side chain size, so it is less likely to affect the function of ORF7a. The alignment of the crystal structures of the ectodomain domains of ORF7a-1 and ORF7a-2 showed that the ectodomains of the two proteins were very similar, with a Cα-RMSD of 0.273. The superposition of the crystal structures revealed that these different amino acids were located on the surface of the protein. They were mainly distributed in the two domains (Figure 5b). One region was in the middle of the three long β-strands; the other region was located in the three loops connecting the β-stands. The loops were spatially close to each other and resembled the antigen-binding region of the antibodies (Figure 5).

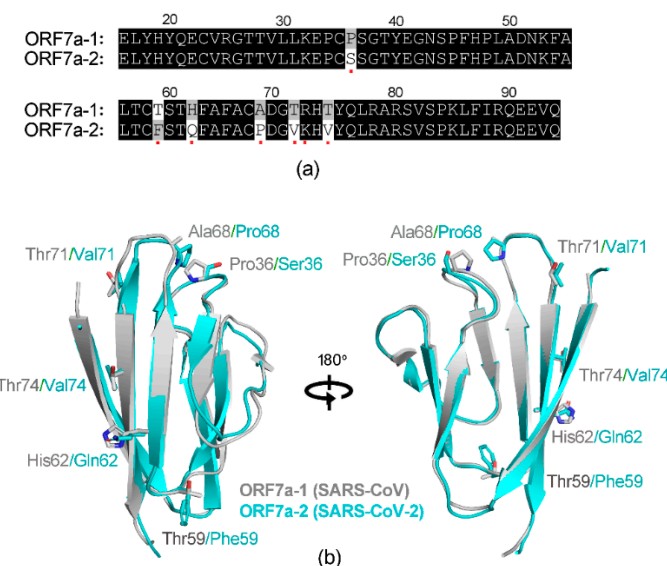

**Figure 5.** Comparative analysis of amino acid sequences and structures of the ectodomains of ORF7a-1 and ORF7a-2. (**a**) Amino acid alignment of the ectodomains of ORF7a-1 and ORF7a-2. Different amino acid residues are indicated by red dots. (**b**) Superposition analysis of structures of the ectodomains of ORF7a-1 and ORF7a-2. Carbon atoms on ORF7a-1 and ORF7a-2 are shown in grey and cyan, respectively. The Protein Data Bank accession numbers of ORF7a-1 and ORF7a-2 are 1XAK and 7CI3, respectively.

## 4. Discussion

Accessory proteins were considered to be important pathogenic factors during SARS-related coronavirus infection [18]. Most of the actions of these accessory proteins were related to the immune system. They either blocked an antiviral response (interferon pathway) such as SARS-CoV/SARS-CoV-2 ORF3b [27,28] and SARS-CoV/SARS-CoV-2 ORF6 [29,30], or they up-regulated host cells cytokine responses, such as SARS-CoV ORF3a and ORF7a [20].

In an earlier study, the transient expression of ORF7a-1 in the A549 cells increased levels of the proinflammatory cytokine IL-8 and the chemokine RANTES (regulated upon activation, normal T cell expressed, and presumably secreted) [20]. ORF7a-2 from SARS-CoV-2 is highly similar to ORF7a-1, and it was, therefore, necessary to verify the role of ORF7a-2 in the immunopathology of SARS-CoV-2. We tried different expression systems, including *E. coli*, *Pichia pastoris*, and *baculovirus*, to produce recombinant ORF7a protein to verify its function in inducing the overproduction of proinflammatory cytokines. Finally, recombinant ORF7a-1 and ORF7a-2 were successfully obtained by refolding the inclusion bodies. Our results confirmed that both ORF7a-1 and ORF7a-2 could significantly up-

regulate the transcription and expression of IL-6, CCL-2, IFN-α, and IFN-β by alveolar epithelial cells. When we prepared our manuscript, several reports on the function of ORF7a were published. A study showed that a large number of cytokines, including IL-1α, IL-1β, IL-6, IL-8, IL-10, TNF-α, and IFN-β, were increased at the transcriptional level when ORF7a-2 was transiently transfected into A549 cells and 16HBE14o cells [21]. In another study, when human peripheral blood mononuclear cells were stimulated with recombinant ORF7a-2, the increased transcriptional levels of IL-6, IL-1β, IL-8, and TNF-α were observed [22].

Based on the above findings, it can be confirmed that ORF7a from SARS-CoV or SARS-CoV-2 promoted cytokine production. It can be concluded that ORF7a mediated the increased cytokine production by both immune cells and lung epithelial cells. It should be noted that the location of ORF7a expression, type of cells, and cytokine detection method adopted in these studies were different. The location of ORF7a expression is actually an important issue to be concerned with. When expressed in cells, ORF7a may be involved in the nuclear factor kappa-B signaling pathway that mediates the transcription of proinflammatory cytokines [21]. In the case of extracellular stimulation with recombinant ORF7a, the interaction target of ORF7a may be immune signaling molecules located on the cell surface. Thus, the mechanisms by which ORF7a up-regulated cytokine production in these two conditions were clearly distinct. In the process of SARS-CoV-2 infection, ORF7a protein appears in both extracellular and intracellular situations. For the intracellular situation, after the virus enters the cell, ORF7a is transcribed and expressed in the cell. For the extracellular situation, ORF7a directly interacts with cell surface molecules, and the signal is transduced into the cell, prompting the transcription of proinflammatory cytokines. The method of cell stimulation with recombinant proteins is equivalent to mimicking virus–cell interactions. Although it had not been experimentally confirmed, ORF7a-2 was likely to be a structural protein on the SARS-CoV-2 particle. ORF7a-1 has been confirmed as a structural protein on SARS-CoV particles by the immunoprecipitation method [31]. ORF7a-2, which had an amino acid sequence identity of up to 86% with ORF7a-1, was probably also the surface membrane protein of the virus.

Identifying the specific protein molecules that interact with ORF7a on the human cell surface will help to explain the molecular mechanisms that up-regulate a range of cytokines. In previous reports, immune cells had often been used as cytokine-overproducing human cells. As early as 2006, the three-dimensional structure of the ectodomain of ORF7a-1 was determined. Researchers found that the ectodomain of ORF7a-1 was similar in structure to intercellular adhesion molecule-1 (ICAM-1) and speculated that ORF7a-1 interacted with lymphocytes. It was proposed that LFA-1, a ligand for ICAM-1, may mediate the interaction of SARS-CoV with immune cells through ORF7a. The interaction of LFA-1 and ORF7a-1 was confirmed experimentally [32,33]. In recent studies, researchers tried to predict the binding mode of ORF7a-2 ectodomain to LFA-1 by computational methods of protein–protein docking and molecular dynamics [34,35]. It was found that the ectodomain domain of ORF7a-2 can efficiently bind CD14$^+$ monocytes, but it cannot be confirmed by in vitro experiments that the binding target of ORF7a-2 on the surface of monocytes is the LFA-1 [22]. Therefore, the specific protein molecule that interacts with ORF7a-2 on immune cells probably was not LFA-1. The alveolar basal epithelial cell A549 used in this study is a non-immune cell, and LFA-1 does not exist on the cell. The specific protein molecule on the alveolar cell that binds to ORF7a needs experimental characterization. The binding target of ORF7a may be a class of immune molecules, such as MHC, that exist on the surface of both immune cells and epithelial cells. It was found that the incubation of ORF7-2 with monocytes severely down-regulated the expression of HLA-DR/DP/DQ molecules [22].

Considering the structure of ORF7, its function in regulating immune responses is not surprising. Accessory proteins ORF7a and ORF8 encoded by SARS-related coronaviruses share the same structural topology, a β-sandwich that consists of seven β-strands. They are the typical topologies of immunoglobulin-like domains [36,37]. It is known that the most important protein molecules in the human immune system contain immunoglobulin-

like domains. These immunoglobulin-like domains play an important role in mediating macromolecular interactions in the immune system. In evolution, viruses can often produce some proteins with immunoglobulin-like domains, disrupting the immune regulation or immune response of host cells, thereby evading clearance by the immune system. ORF8 is a well-studied accessory protein of SARS-CoV-2. Its mechanism of action was well explained by its location and target molecule on the cell surface. ORF8 can be secreted extracellularly to become an enhanced version of IL-17 and induce a significantly stronger inflammatory response than the host IL-17 [38]. ORF8 can bind to monocytes and NK cells to evade the immune system by reducing the ability of the infected cells to have antibody-dependent cytotoxicity [39].

The present study is the first to analyze the differences in the ability of ORF7a-1 and ORF7a-2 to regulate cytokine production. Previous studies have found that ORF7a-2 ectodomain and ORF7a-1 ectodomain have significantly different binding abilities to CD14$^+$ monocytes (the former is much stronger than the latter), which implied the existence of a potential difference between these two highly conserved accessory proteins [22]. Among the different amino acid residues between ectodomains of ORF7a-1 and ORF7a-2, residue 72 is an amino acid with similar properties and side chain size, so it is less likely to affect the function of ORF7a. It can be speculated that three amino acid variations (P36S, A68P, T71V) on the loops of the ectodomains of the two ORF7a were the main factors for their functional differences.

Many reports, including this study, provided evidence for the association of accessory proteins with the immunopathology of SARS-CoV-2. However, it should be noted that these conclusions were based on experiments with a single accessory protein, while interactions between viral proteins were not taken into account. More importantly, the expression of accessory proteins in the experiment was likely to be much higher than the actual situation of virus infection. Therefore, it is necessary to validate the functions of these accessory proteins in laboratories that are qualified to carry out genetic modification of SARS-CoV-2. Since the outbreak of SARS-CoV-2, a number of natural mutations/deletions of accessory proteins have been found around the world. Some of the mutations have been linked to the symptoms and infection process of SARS-CoV-2. A typical example was the accessory protein ORF8. Infection with SARS-CoV-2 carrying an ORF8 mutant (D382 deletion or L84S mutation) resulted in a milder inflammatory response and attenuated disease outcomes [40]. ORF7a mutants have also been reported. ORF7a mutants missing the C-terminal half led to SARS-CoV-2 replication defects and easy clearance by the immune system [41]. Similarly, viral genome sequencing and clinical data showed that the ORF7a mutant A105V was associated with increased severity and lethality in a cohort of Romanian patients [42]. These mutants provide important evidence for identifying the roles of accessory proteins in the pathogenic mechanism of SARS-CoV-2.

**Supplementary Materials:** The following supporting information can be downloaded at: https://www.mdpi.com/article/10.3390/covid2100104/s1, Figure S1: Scheme of DNA plasmid used for production of ectodomains of ORF7a-1 and ORF7a-2.

**Author Contributions:** Methodology, D.C.; conceptualization, Z.Z.; writing—original draft preparation, Z.H.; funding acquisition, Z.H. All authors have read and agreed to the published version of the manuscript.

**Funding:** This research was funded by Hubei Provincial Natural Science Foundation of China, grant number 2020CFB803.

**Institutional Review Board Statement:** Not applicable.

**Informed Consent Statement:** Not applicable.

**Data Availability Statement:** Not applicable.

**Conflicts of Interest:** The authors declare no conflict of interest.

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
