# Peer review of "Expression and Purification of Recombinant SARS-CoV-2 Accessory Protein ORF7a and Functional Analysis of Its Role in Up-Regulating Cytokine Production"

_covid, doi:10.3390/covid2100104_

Round 1
Reviewer 1 Report
The manuscript by Chen, Zheng and Han, entitled “Recombinant Expression of SARS-CoV-2 Accessory Protein 2 ORF7a and Functional Analysis Its Role in Up-Regulating Cytokine Production” describes the heterologous expression in E. coli of the SARS-CoV and SARS-CoV-2 recombinant proteins ORF7a, their purification after inclusion bodies denaturation and protein refolding, and the characterization of their ability to induce the up-regulation of cytokines in A549 cells when added extracellularly to the supernatant of the mammalian cell culture.
Overall, the work is scientifically sound and methodologically well-conceived, and the results shown by the authors are consistent with their hypothesis and drawn conclusions. With this in mind, it is opinion of this reviewer that the work would be informative and interesting for the readership of COVID, and should be therefore published.
However, major concerns lie in the fact that some introductory concepts and considerations made by the authors in the discussion, would better contextualize the research if placed in the introduction. In addition, the manuscript would benefit in clarity if i) the choice of the experimental strategy would be explained first to the reader, and ii) the structural comparison of the two proteins would be moved from the discussion to the results, as this analysis was anticipated by the authors themselves at the end of the introduction as something undertaken by them in this study in light of their experimental findings.
Finally, the manuscript style and format would benefit from a thorough check by the authors of the English spell, use of italics, spaces between digits and their units and – above all – of the consistent acronymization of all scientific terms.
This reviewer encourages the authors to take a look at the major and minor suggestions below, which are intended to help highlighting the quality of the research work that they have performed.
Major suggestions:
Title: consider re-phrasing as follows “Expression and purification of recombinant SARS-CoV-2 accessory protein ORF7a and functional analysis of its role in up-regulating cytokine production”
Line 75 and lines 289-302: a reader who is not familiar with the molecular biology of coronaviruses and with that of SARS-CoV-s proteins, may not know that ORF7a proteins have an ectodomain. Not even, in general, that the protein is an immunoglobulin-like protein. Please, the authors should dedicate one or two phrases corroborated by appropriate references on the description of the ORF7a structural organization. Moreover, authors have already provided such description to the readers, however confining it at the end of the manuscript, in the discussion session. This reviewer encourages them to move that part to the end of the introduction, for more clarity.
Lines 308-323: all this part, including Figure 5, should become a new paragraph in the results section. In fact, the authors state at lines 79-80 (end of the introduction) that they have "explored functional differences through three-dimensional structural analysis" and therefore such analysis cannot be postponed to the discussion. Furthermore, here the work by Zhou et al., 2021 (iScience, reference 32) should be properly cited to introduce the SARS-CoV-2 ORF7a crystal structure (PDB: 7IC3) as it was properly cited elsewhere to discuss functional aspects. Please, when making this new paragraph of the results (e.g. entitled “structural analysis of …”) consider to also add a corresponding brief session in the methods (e.g. entitled “molecular graphics”) in which the authors should explain the molecular viewer software used and the tools used to superimpose the two structures (e.g. PyMOL? Chimera? Match-Align, TM-align tools?). Please, add in the results some metrics for objective structure comparison (e.g. RMSD, over how many residues it was calculated, SDM and/or Q-score, TM-score etc.). Finally, consider adding an amino acid sequence alignment highlighting the residues described in the text.
Minor suggestions:
Line 43-44: Please, remove "generally refers [...] cytokines. It", as it is a redundant phrase. Write instead directly "CRS has attracted..."
Line 45: "severe infections by respiratory viruses such as... "
Line 46-48; simplify like this, "in controlling the infection and limiting the spread of viruses [8]".
Line 51: please, the authors are encouraged to write each acronym in its explicit form first, followed by the corresponding acronym in round brackets, and from that point onward in the paper to use only the acronym. Specifically, please, write here: "cytokines and chemokines such as tumor necrosis factor alpha (TNF-?), interleukin-1 beta (IL-1?), IL-6, IL-8 and IL17, in severe... [...]"
Line 70: identical in amino acid sequence
Lines 72-73: please, rephrase as follows "[...] in amino acid sequence between the ORF7a encoded by the two viruses, it is likely that also ORF7a-2 has the capability to stimulate host cells..."
Line 76: "highly-pure recombinant ORF7a [...]"
Line 77: "[...] it was demonstrated that both ORF7a-1 and ORF7a-2 have the ability [...]"
Line 78: "albeit with differences."
Line 83: "SARS-CoV ORF7a-1 and SARS-CoV-2 ORF7a-2"
Line 86: "cells were preserved"
Line 87: DMEM medium (add brand) containing 10 % (space between digits and unit) metal bovine serum (brand) at 37 ºC (space) and 5 % (space)
Line 92: "Gene fragments encoding for the ectodomains"; also, please, add brand of purchased restriction enzymes and E. colicells
Line 93: add acronym (E. coli)
Lines 97-107: rpm; 37 ºC ;4 ºC (with space)
Line 109: which nickel column? Please, provide type of resin and brand
Lines 111-112: please, remove "final" and substitute "obtained" with "purified".
Line 116: please, invert; first the explicit form, then its acronym (SDS-PAGE)
Line 119: polyvinylidene fluoride (PVDF) membrane
Line 122: "[...] ECL detection system (brand)"
Line 125: cells (brand)
Line 126: total amount of protein is only partially informative. Could the authors perhaps provide some indication about final concentration and/or volume of cell-culture supernatant?
Line 131: brand of TRIzol
Line 136-140: detection system (brand); software (brand); please, insert explicit form (GAPDH); method [add reference]
Line 139: each cytokine? The reviewer guesses are the same enlisted as acronyms at line 195. Please, authors should add here all the cytokine names, with their acronyms (only if they were mentioned above in the text) or with their explicit form plus the acronym (e.g. interferon alpha?) for those not mentioned before. Then, at line 195 below, only the acronyms can be mentioned.
Line 143: rpm
Line 145: since the authors kindly provided to readers a reference for the bacterial induction method (Studier, 2005) for consistency would be great if they also add a reference for the double-antibody sandwich one.
Line 156: results indicate that [...] are successfully expressed
Line 157: "ectodomains [...] were produced in the non-soluble form..." E. coli in italic
Line 160: please remove brackets; denaturant buffer containing...
Line 161: please, remove method and substitute with "dropwise dilution".
Line 162-164: (Figure 1, lane 3); (Figure 1, lane 4)
Line 176: but also to concentrate;
Line 179: Figure 1, lane 7
Line 183: Please, add ORF7 in the caption
Line 185: "A protein band (please, remove space) between..."
Line 201: why "not shown"? If it is important to mention, although for some reason it could not be shown, the authors could maybe kindly provide a brief explanation for not showing this.
Line 225: please, explicit form of RANTES
Line 226: "it was therefore necessary to..."
Line 233: please remove (Figure 3), since we are in the discussion section.
Line 243-247: all this part should be moved to the end of the introduction, in order to prepare the readers and let them understand the value and the importance of the experimental strategy adopted by the authors.
Line 250: "to be concerned of" or, "of concern".
Line 251: explicit form of NF-kB
Line 304: please, remove (Figure 3) from discussion
Line 324: please, remove "although"
Author Response
The manuscript by Chen, Zheng and Han, entitled “Recombinant Expression of SARS-CoV-2 Accessory Protein 2 ORF7a and Functional Analysis Its Role in Up-Regulating Cytokine Production” describes the heterologous expression in E. coli of the SARS-CoV and SARS-CoV-2 recombinant proteins ORF7a, their purification after inclusion bodies denaturation and protein refolding, and the characterization of their ability to induce the up-regulation of cytokines in A549 cells when added extracellularly to the supernatant of the mammalian cell culture.
Overall, the work is scientifically sound and methodologically well-conceived, and the results shown by the authors are consistent with their hypothesis and drawn conclusions. With this in mind, it is opinion of this reviewer that the work would be informative and interesting for the readership of COVID, and should be therefore published.
However, major concerns lie in the fact that some introductory concepts and considerations made by the authors in the discussion, would better contextualize the research if placed in the introduction. In addition, the manuscript would benefit in clarity if i) the choice of the experimental strategy would be explained first to the reader, and ii) the structural comparison of the two proteins would be moved from the discussion to the results, as this analysis was anticipated by the authors themselves at the end of the introduction as something undertaken by them in this study in light of their experimental findings.
Response: We had moved the comparative analysis of protein structure to the section of experimental results, and added the corresponding experimental methods in Materials and Methods.
Finally, the manuscript style and format would benefit from a thorough check by the authors of the English spell, use of italics, spaces between digits and their units and – above all – of the consistent acronymization of all scientific terms.
Response:During the revision of the manuscript, we had done our best to remove language and formatting errors in the manuscript.
This reviewer encourages the authors to take a look at the major and minor suggestions below, which are intended to help highlighting the quality of the research work that they have performed.
Major suggestions:
Title: consider re-phrasing as follows “Expression and purification of recombinant SARS-CoV-2 accessory protein ORF7a and functional analysis of its role in up-regulating cytokine production”
Response: We think that the article title suggested by the reviewer is better. We adopt this new title.
Line 75 and lines 289-302: a reader who is not familiar with the molecular biology of coronaviruses and with that of SARS-CoV-s proteins, may not know that ORF7a proteins have an ectodomain. Not even, in general, that the protein is an immunoglobulin-like protein. Please, the authors should dedicate one or two phrases corroborated by appropriate references on the description of the ORF7a structural organization. Moreover, authors have already provided such description to the readers, however confining it at the end of the manuscript, in the discussion session. This reviewer encourages them to move that part to the end of the introduction, for more clarity.
Response: We had added a description of the ORF7a domain composition and related references to the last paragraph of the introduction section. This allows the reader easier to accept ORF7a ectodomain described in in Materials and Methods.
Lines 308-323: all this part, including Figure 5, should become a new paragraph in the results section. In fact, the authors state at lines 79-80 (end of the introduction) that they have "explored functional differences through three-dimensional structural analysis" and therefore such analysis cannot be postponed to the discussion. Furthermore, here the work by Zhou et al., 2021 (iScience, reference 32) should be properly cited to introduce the SARS-CoV-2 ORF7a crystal structure (PDB: 7IC3) as it was properly cited elsewhere to discuss functional aspects. Please, when making this new paragraph of the results (e.g. entitled “structural analysis of …”) consider to also add a corresponding brief session in the methods (e.g. entitled “molecular graphics”) in which the authors should explain the molecular viewer software used and the tools used to superimpose the two structures (e.g. PyMOL? Chimera? Match-Align, TM-align tools?). Please, add in the results some metrics for objective structure comparison (e.g. RMSD, over how many residues it was calculated, SDM and/or Q-score, TM-score etc.). Finally, consider adding an amino acid sequence alignment highlighting the residues described in the text.
Response: We think it was a good suggestion to place the structural alignment analysis as an independent experimental result placed the Results section. In the revised manuscript, we had followed this suggestion. At the same time, according to the reviewer's suggestion, we also added pictures of protein amino acid sequence alignment in the Results section. Structural alignment and amino acid sequence alignment are presented in same figure (Figure 5). Corresponding content has been added to the Materials and Methods section. The caption of Figure 5 was also modified accordingly.
Minor suggestions:
Line 43-44: Please, remove "generally refers [...] cytokines. It", as it is a redundant phrase. Write instead directly "CRS has attracted..."
Response: We agree with this suggestion. In the revised manuscript, we had followed this suggestion.
Line 45: "severe infections by respiratory viruses such as... "
Response: We had corrected the sentence as that suggested.
Line 46-48; simplify like this, "in controlling the infection and limiting the spread of viruses [8]".
Response: We agree with this suggestion. In the revised manuscript, we had followed this suggestion.
Line 51: please, the authors are encouraged to write each acronym in its explicit form first, followed by the corresponding acronym in round brackets, and from that point onward in the paper to use only the acronym. Specifically, please, write here: "cytokines and chemokines such as tumor necrosis factor alpha (TNF-?), interleukin-1 beta (IL-1?), IL-6, IL-8 and IL17, in severe... [...]"
Response: We had followed this suggestion and annotated the full and abbreviated forms of cytokines that first appeared in the manuscript.
Line 70: identical in amino acid sequence
Response: We had replaced “protein” with “amino acid”.
Lines 72-73: please, rephrase as follows "[...] in amino acid sequence between the ORF7a encoded by the two viruses, it is likely that also ORF7a-2 has the capability to stimulate host cells..."
Response: We had rephrased the sentence according to the suggestion by reviewer.
Line 76: "highly-pure recombinant ORF7a [...]"
Response: We had corrected the mistake.
Line 77: "[...] it was demonstrated that both ORF7a-1 and ORF7a-2 have the ability [...]"
Response: We had rephrased the sentence according to the suggestion by reviewer.
Line 78: "albeit with differences."
Response: We had rephrased the sentence according to the excellent expression by reviewer.
Line 83: "SARS-CoV ORF7a-1 and SARS-CoV-2 ORF7a-2"
Response: We had added in the information.
Line 86: "cells were preserved"
Response: We had corrected the mistake.
Line 87: DMEM medium (add brand) containing 10 % (space between digits and unit) metal bovine serum (brand) at 37 ºC (space) and 5 % (space)
Response: We had added products information and the spaces.
Line 92: "Gene fragments encoding for the ectodomains"; also, please, add brand of purchased restriction enzymes and E. colicells
Response: We had added brand information of restriction enzymes and cell line.
Line 93: add acronym (E. coli)
Response: We had corrected the mistake.
Lines 97-107: rpm; 37 ºC ;4 ºC (with space)
Response: These spaces had been added and unified the format units and numbers in the manuscript.
Line 109: which nickel column? Please, provide type of resin and brand
Response: The information of column was added.
Lines 111-112: please, remove "final" and substitute "obtained" with "purified".
Response: The suggested corrections have been made.
Line 116: please, invert; first the explicit form, then its acronym (SDS-PAGE)
Response: The text was modified.
Line 119: polyvinylidene fluoride (PVDF) membrane
Response: Full name of PVDF had been added.
Line 122: "[...] ECL detection system (brand)"
Response: The information was added
Line 125: cells (brand)
Response: The information was added
Line 126: total amount of protein is only partially informative. Could the authors perhaps provide some indication about final concentration and/or volume of cell-culture supernatant?
Response: Volume of cell-culture supernatant was provided. Protein concentration was seen in the Result section (3.1).
Line 131: brand of TRIzol
Response: The information was added
Line 136-140: detection system (brand); software (brand); please, insert explicit form (GAPDH); method [add reference]
Response: The information and reference were added
Line 139: each cytokine? The reviewer guesses are the same enlisted as acronyms at line 195. Please, authors should add here all the cytokine names, with their acronyms (only if they were mentioned above in the text) or with their explicit form plus the acronym (e.g. interferon alpha?) for those not mentioned before. Then, at line 195 below, only the acronyms can be mentioned.
Response: All the cytokines that were quantified were listed in the text.
Line 143: rpm
Response: ”rpm” was used.
Line 145: since the authors kindly provided to readers a reference for the bacterial induction method (Studier, 2005) for consistency would be great if they also add a reference for the double-antibody sandwich one.
Response: Reference for the double-antibody sandwich method was added as reference 26.
Line 156: results indicate that [...] are successfully expressed
Response: The mistake had been corrected.
Line 157: "ectodomains [...] were produced in the non-soluble form..." E. coli in italic
Response: The mistakes had been corrected.
Line 160: please remove brackets; denaturant buffer containing...
Response: The mistakes had been corrected.
Line 161: please, remove method and substitute with "dropwise dilution".
Response: The better representation from the reviewers was adopted.
Line 162-164: (Figure 1, lane 3); (Figure 1, lane 4)
Response: The suggestion was adopted.
Line 176: but also to concentrate;
Response: The suggestion was adopted.
Line 179: Figure 1, lane 7
Response: The suggestion was adopted.
Line 183: Please, add ORF7 in the caption
Response: The mistakes had been corrected.
Line 185: "A protein band (please, remove space) between..."
Response: The mistakes had been corrected.
Line 201: why "not shown"? If it is important to mention, although for some reason it could not be shown, the authors could maybe kindly provide a brief explanation for not showing this.
Response: “not” was missed in the sentence. It is “CXCL-9 was not detected”.
Line 225: please, explicit form of RANTES
Response: The explicit form was used.
Line 226: "it was therefore necessary to..."
Response: The sentence was modified.
Line 233: please remove (Figure 3), since we are in the discussion section.
Response: It had been removed.
Line 243-247: all this part should be moved to the end of the introduction, in order to prepare the readers and let them understand the value and the importance of the experimental strategy adopted by the authors.
Response: The description of our unique experimental design had been moved from Discussion to the Introduction.
Line 250: "to be concerned of" or, "of concern".
Response: The sentence had been corrected.
Line 251: explicit form of NF-kB
Response: The explicit form of NF-kB was used.
Line 304: please, remove (Figure 3) from discussion
Response: It had been removed.
Line 324: please, remove "although"
Response: It had been removed.

Reviewer 2 Report
The work addresses what can be a useful piece of information on an individual accessory protein (ORF7a) of both SARS-CoV and SARS-CoV2: its in vitro capacity to up-regulate pro-inflammatory cytokines. Due to several minor and medium issues (mainly involving methodologies and statistics), I don't see this manuscript ready for publication the way it's currently presented. Please see my comments below for adjusting it accordingly.
lines 92-94: Authors must provide a supplementary figure on the cassette scheme for the heterologous expression, depicting the plasmid vector (pET-28a), position of the restrictions sites for the insert, and the insert sequence itself flanked by the primers used in the PCR reaction performed prior to cloning. Finally, The genome accession number for the strains authors are relying on must also be provided.
line 156: Replace "express" by "expressed".
line 159: Replace "was" by "were". As far as I understood, it's referring to "centrifugation pellets" (plural), isn't it?
line 183: Figure 2 caption needs adjusment.
lines 194-205: Authors must inform which statistical test was employed, as well as the p-values of the comparisons emphasized in that paragraph. As usually observed in many papers, asterisk(s) could also be used on Figure 3 to indicate p-values below certain threshold(s) observed mainly in significant differences.
lines 197-198: That sentence needs adjustment for a better reading. Replace "stronger" by "more effective", and "promoting" by "up-regulating".
lines 208-216: Same statistical concern raised above (for lines 194-205) also applies to this paragraph and Figure 4.
line 301: Replace "evade immune evasion" by "evade immune system".
line 307: Replace "highly homologous" by "highly conserved".
line 308: Replace "ectodomain" by "ectodomains".
lines 321-322: More information must be provided about the methods for the visualization of those ORF7a 3D structuresa on Figure 5 (e.g. tool and its respective settings used). This info must be added on both Figure caption and in the Methods section.
Author Response
The work addresses what can be a useful piece of information on an individual accessory protein (ORF7a) of both SARS-CoV and SARS-CoV2: its in vitro capacity to up-regulate pro-inflammatory cytokines. Due to several minor and medium issues (mainly involving methodologies and statistics), I don't see this manuscript ready for publication the way it's currently presented. Please see my comments below for adjusting it accordingly.
lines 92-94: Authors must provide a supplementary figure on the cassette scheme for the heterologous expression, depicting the plasmid vector (pET-28a), position of the restrictions sites for the insert, and the insert sequence itself flanked by the primers used in the PCR reaction performed prior to cloning. Finally, The genome accession number for the strains authors are relying on must also be provided.
Response: A supplementary figure had been prepared to provide information for gene cloning.
line 156: Replace "express" by "expressed".
Response: The mistake had been corrected.
line 159: Replace "was" by "were". As far as I understood, it's referring to "centrifugation pellets" (plural), isn't it?
Response: The mistake had been corrected.
line 183: Figure 2 caption needs adjusment.
Response: We had carefully checked the caption.
lines 194-205: Authors must inform which statistical test was employed, as well as the p-values of the comparisons emphasized in that paragraph. As usually observed in many papers, asterisk(s) could also be used on Figure 3 to indicate p-values below certain threshold(s) observed mainly in significant differences.
Response: We had applied the P-value statistics (labeled with asterisk) for Figure 3 and 4. The corresponding significant differences were addressed in the paragraph.
lines 197-198: That sentence needs adjustment for a better reading. Replace "stronger" by "more effective", and "promoting" by "up-regulating".
Response: These suggestions had been adopted.
lines 208-216: Same statistical concern raised above (for lines 194-205) also applies to this paragraph and Figure 4.
Response: We had applied the P-value statistics (labeled with asterisk) for Figure 3 and 4. The corresponding significant differences were addressed in the paragraph.
line 301: Replace "evade immune evasion" by "evade immune system".
Response: The error had been corrected.
line 307: Replace "highly homologous" by "highly conserved".
Response: These suggestions had been adopted.
line 308: Replace "ectodomain" by "ectodomains".
Response: The error had been corrected.
lines 321-322: More information must be provided about the methods for the visualization of those ORF7a 3D structuresa on Figure 5 (e.g. tool and its respective settings used). This info must be added on both Figure caption and in the Methods section.
Response: Figure 5 had been moved to the Result section. The corresponding methods to perform structural alignment and generate the figure had been addressed in the Material and Methods section.

Reviewer 3 Report
Dear Editor,
Chen et al have presented a detailed study on the characterization and role of SARS-CoV-2 ORF7a protein in upregulating cytokine production. The authors use recombinant ORF7a protein from two closely related SARS coronaviruses to examine expression levels of cytokines in A549 cells. The authors conclude that ORF7a is an important accessory protein and immune regulator in the pathogenic nature of SARS-CoV-2.
The authors present a comprehensive introduction, discussion and conclusion in a clear and concise manner. I would also recommend the authors to proofread the manuscript for better readability. However, I do have few minor suggestions with the methods and results for better clarity that the authors could briefly elaborate on. These are outlined below
1. Line 75: Indicate N terminal ectodomains for ORF7A proteins
2. Line 86: Please provide citation for A549 cells (Girard et al 1973 J nat cancer Institute)
3. It would be useful to know if the authors tested other cell lines like NCI H441, immortalized AT2 cells.
4. It would be useful to describe methods in detail for example: Line 109: Please indicate the concentration of imidazole used for elution, how much ORF7a-1 was used for immunoblotting, dilution of ORF7a-2 polyclonal antibody used.
5. For cell experiments, it would be useful to know how the authors determined how many cells and how much protein to use. Did the authors test different dilutions/titrations of protein?
6. Have the authors tested transcription of cytokines when you do eukaryotic expression of ORF7a-1/ORF7b-1. In HEK293 cells it was shown to inhibit cell growth. Is there any evidence in A549 cells?
7. For RT-qPCR, how many replicates were tested? It would be useful to include statistical analysis like T-test to show if there is a significant difference in expression. It is hard to conclude with the data shown
8. Line 260: Is there evidence for stimulation of cells with recombinant proteins. Please provide citations and examples.
9. Figure 5: Please include an amino acid alignment for ORF7a-1 and ORF7a-2 to show amino acid differences.
Author Response
Chen et al have presented a detailed study on the characterization and role of SARS-CoV-2 ORF7a protein in upregulating cytokine production. The authors use recombinant ORF7a protein from two closely related SARS coronaviruses to examine expression levels of cytokines in A549 cells. The authors conclude that ORF7a is an important accessory protein and immune regulator in the pathogenic nature of SARS-CoV-2.
The authors present a comprehensive introduction, discussion and conclusion in a clear and concise manner. I would also recommend the authors to proofread the manuscript for better readability. However, I do have few minor suggestions with the methods and results for better clarity that the authors could briefly elaborate on. These are outlined below
- Line 75: Indicate N terminal ectodomains for ORF7A proteins
Response: The suggestion had been adopted.
- Line 86: Please provide citation for A549 cells (Girard et al 1973 J nat cancer Institute)
Response: The paper had been cited as reference 23.
- It would be useful to know if the authors tested other cell lines like NCI H441, immortalized AT2 cells.
Response: It would be interesting to test ORF7A on other similar lung cells, such as the alveolar carcinoma cell NCI H441 like A549 and the lung epithelial stem cell AT2. But we don't have the experimental condition right now to extend our studies to these cells.
- It would be useful to describe methods in detail for example: Line 109: Please indicate the concentration of imidazole used for elution, how much ORF7a-1 was used for immunoblotting, dilution of ORF7a-2 polyclonal antibody used.
Response: The imidazole concentration and elution method were provided in the paragraph. The approximate amounts of recombinant proteins and polyclonal antibody were indicated in the corresponding positions in the text.
- For cell experiments, it would be useful to know how the authors determined how many cells and how much protein to use. Did the authors test different dilutions/titrations of protein?
Response: We determined the cell density by microscopy and performed the experiment when the cells grew to 80% density. The amount of recombinant protein used had been indicated in the text. We tried different concentrations of recombinant proteins but just evaluated the effect of protein concentrations on cell growth. The final used protein concentration was a relatively higher concentration that did not affect cell growth. We also attempted to find a reference in literature of similar experiments, and in a study that also used recombinant ORF7a protein for cell stimulation, the researchers did not explicitly provide protein concentrations. In the Discussion section of this manuscript, we also mentioned that the protocol used in the study may not match the protein concentration during the viral infection and may be much higher.
- Have the authors tested transcription of cytokines when you do eukaryotic expression of ORF7a-1/ORF7b-1. In HEK293 cells it was shown to inhibit cell growth. Is there any evidence in A549 cells?
Response: We did not use mammalian cells for recombinant expression and purification to prepare recombinant ORF7a protein, nor is such an experimental protocol seen in the published study. We have seen multiple studies in which researchers had used transient expression to express ORF7a in mammalian cells to measure cytokine transcription. We did not take this approach.
- For RT-qPCR, how many replicates were tested? It would be useful to include statistical analysis like T-test to show if there is a significant difference in expression. It is hard to conclude with the data shown
Response: In experiments to measure cytokine transcription by qPCR, we performed three qPCR replicates per cell sample (cDNA). In the revised manuscript, we used P-values to analyze the significance of the differences between the experimental groups, and marked them in the pictures in the usual way of asterisks.
- Line 260: Is there evidence for stimulation of cells with recombinant proteins. Please provide citations and examples.
Response: In reference 22 of the manuscript, researchers stimulated peripheral blood mononuclear cells extracellularly with recombinant ORF7a protein. We cited the article in the Discussion and pointed out that the difference between this study and ours is that we used lung epithelial cells, they used immune cells.
- Figure 5: Please include an amino acid alignment for ORF7a-1 and ORF7a-2 to show amino acid differences.
Response: We had prepared a new Figure 5. Figure 5a is the amino acid alignment of ORF7a-1 and ORF7a-2.
